# The Role of Lifestyle Interventions in the Prevention and Treatment of Gestational Diabetes Mellitus

**DOI:** 10.3390/medicina59020287

**Published:** 2023-02-01

**Authors:** Hala Zakaria, Salah Abusanana, Bashair M. Mussa, Ayesha S. Al Dhaheri, Lily Stojanovska, Maysm N. Mohamad, Sheima T. Saleh, Habiba I. Ali, Leila Cheikh Ismail

**Affiliations:** 1Clinical Science Department, College of Medicine, University of Sharjah, Sharjah 27272, United Arab Emirates; 2Diabetes and Endocrinology Department, University Hospital Sharjah, Sharjah 27272, United Arab Emirates; 3Department of Nutrition and Health, College of Medicine and Health Sciences, United Arab Emirates University, Al Ain 15551, United Arab Emirates; 4Institute for Health and Sport, Victoria University, Melbourne, VIC 3011, Australia; 5Department of Clinical Nutrition and Dietetics, College of Health Sciences, University of Sharjah, Sharjah 27272, United Arab Emirates; 6Nuffield Department of Women’s & Reproductive Health, University of Oxford, Oxford OX1 2JD, UK

**Keywords:** gestational diabetes mellitus, GDM, lifestyle interventions, prevention, diet, physical activity

## Abstract

Gestational diabetes mellitus (GDM) is one of the most common pregnancy-related endocrinopathies, affecting up to 25% of pregnancies globally. GDM increases the risk of perinatal and delivery complications, and the chance of developing type 2 diabetes mellitus and its complications, including cardiovascular diseases. This elevated risk is then passed on to the next generation, creating a cycle of metabolic dysfunction across generations. For many years, GDM preventive measures have had inconsistent results, but recent systematic reviews and meta-analyses have identified promising new preventative routes. This review aims to summarize the evidence investigating the efficacy of lifestyle treatments for the prevention of GDM and to summarize the effects of two lifestyle interventions, including physical activity and dietary interventions. Based on the present research, future studies should be conducted to investigate whether initiating lifestyle interventions during the preconception period is more beneficial in preventing GDM. In addition, research targeting pregnancy should be designed with a personalized approach. Therefore, studies should customize intervention approaches depending on the presence of modifiable and non-modifiable risk factors at the individual level.

## 1. Introduction

Gestational diabetes (GDM) is a condition defined as hyperglycemia due to an increase in placental hormone secretion during pregnancy. GDM occurs when the pancreas secretes insufficient amounts of insulin to handle the rise in blood sugar and is usually detected by 13–26 gestational weeks or early in the third trimester of pregnancy [1]. Although GDM may recuperate after delivery, it has been linked to severe maternal outcomes, including type 2 diabetes mellitus (T2DM) [2], cardiovascular disease [3], and metabolic syndrome [4,5,6]. Despite the maternal outcomes, the offspring of GDM pregnancies had a higher frequency of childhood overweight/obesity [7], dysglycemia, dyslipidemia, and metabolic syndrome [8]. Moreover, there is an increased risk of developing T2DM in childhood and neurodevelopmental impairments [9,10]. This proclivity for metabolic dysfunction that connects mother and child may be the result of a combination of factors, including: (i) shared genetic factors, (ii) the home environment and associated lifestyle, and (iii) fetal exposure to the altered intrauterine environment of the GDM pregnancy, which may program adverse developmental pathways according to the Developmental Origins of Health and Disease paradigm [11].

The development of GDM is associated with modifiable and non-modifiable risk factors. Non-modifiable risk factors include genetic inherent risk factors [10], race (Hispanic or Asian) [12], maternal age, first-degree relatives with T2DM, and a history of Polycystic Ovarian Syndrome (PCOS) [13]. Alternatively, lifestyle and environmental factors are among the modifiable risk factors that have been shown to influence the development of GDM. These can be divided into pre-pregnancy and peri-pregnancy risk factors. The former is mostly lifestyle-related, including pre-pregnancy obesity and overweight, poor diet quality, and physical inactivity [14], while the latter for the most part denotes excessive weight gain during pregnancy, which is defined as gestational weight gain (GWG) [15].

Various variables influence maternal glucose control, including, physiological changes during pregnancy, pathological circumstances, and maternal nutrition. The kinds of carbohydrates in the diet have a significant impact on maternal glucose via their direct effect on glycemia [16]. Carbohydrate type and glycemic index (GI) of the diet, either encourage or prevent excessive hyperglycemia during pregnancy caused by pathological circumstances, or the mother’s incapacity to cope with the physiological Insulin Resistance (IR) of pregnancy [16]. Furthermore, physical activity (PA) has been demonstrated to enhance glycemic management, in part through enhancing contraction-mediated glucose absorption into skeletal muscle [17].

After delivery, the IR of pregnancy subsides, and blood glucose levels in women with GDM often recover to normal, obviating the need for continuous glucose-lowering medication [11]. While the documented temporary hyperglycemia during pregnancy may reinforce the impression of GDM as a strictly medical consequence of pregnancy, the glycemic ramifications of this diagnosis extend well beyond pregnancy [11]. The beta-cell dysfunction that results in poor compensation for pregnancy IR is both chronic and progressive in nature [11]. As a result, women who acquire GDM often have progressive decrease in beta-cell function in the years after the index pregnancy, resulting in increased glycemia over time, which can lead to pre-diabetes and T2DM [11]. This review aims to summarize the evidence investigating the efficacy of lifestyle treatments for the prevention of GDM and to summarize the effects of two lifestyle interventions, including physical activity and dietary interventions.

## 2. Methodology

A comprehensive literature search was performed using PubMed, Cochrane Reviews, Google Scholar, SpringerLink, and Scopus database platforms for relevant articles. Articles that were published in the past 10 years, from 1 July 2012, to 1 July 2022, were considered in the review. The search terms used included ‘gestational diabetes’ and ‘lifestyle interventions’, or ‘physical activity’, or ‘dietary interventions and ‘prevention’, were included in the search. The articles obtained from the search were assessed for topic relevance and hand-reviewed to further find related publications. Randomized controlled trials (RCTs), meta-analyses, systematic reviews, and cohort studies were included. Only articles that involved interventions for adults (18 and older) randomized and non-randomized studies involving intervention groups (with diet and/or physical activity) and control groups (without intervention or standard treatment/placebo) were also included. Meta-analyses that included RCTs that measured the effect of physical activity, dietary interventions, combined lifestyle interventions, and supplements on the prevention of GDM were included. In total, 1284 articles were identified in the initial search. After the removal of 299 duplicates, 985 titles or abstracts were screened, of which 840 were not relevant and were excluded. The remaining 145 full-text articles were further reviewed for eligibility, leading to the elimination of 115 full-text articles due to the inclusion criteria, as outlined in Figure 1. Therefore, 30 articles were included in this review.

## 3. Current Physical Activity Recommendations in GDM Treatment

For women with GDM, PA is a vital component of lifestyle modification. Exercise improves glycemic control by increasing insulin sensitivity through improved muscle glucose uptake as well as preventing excessive GWG, with beneficial implications for fetal development and future offspring’s health. Recent literature has shown significant inverse associations between exercise and numerous glycemic control indicators when exercise training is utilized as an “adjunct” treatment [18,19]. The American Diabetes Association (ADA) recommends a minimum of 20–50 min/day, 2–7 days/week, of moderate intensity PA (aerobic, resistance, or both) [20]. A review of the international recommendations from different guidelines concluded that for patients with GDM, an aerobic training (walking, jogging, running, elliptical machine, cycling, swimming, and aqua aerobics) for at least 3–4 times/week of 50–150 min/week, with an upper limit of 30 min per day and at least 2 times of resistance training per week, is the most effective PA intervention for GDM as the first line-treatment, along with dietary modifications [21].

## 4. Effect of Physical Activity on Prevention of GDM

A myriad of meta-analyses on the impact of PA in preventing GDM pre-pregnancy, early in pregnancy, and during pregnancy, have been published since 2015 [22,23,24,25,26,27,28,29,30,31]. All meta-analyses except that of Madhuvrata et al. [31], revealed that PA protects against the development of GDM with a strong inverse relationship and an overall risk reduction ranging from 23–59% [22,31]. Table 1 summarizes the results of these meta-analyses. These findings are comparable to those of other reviews, including the United States PA recommendations, which revealed “strong” evidence of an inverse association between PA and the risk of GDM [32,33].

A meta-analysis, conducted by Ying Yu et al., measured the effect of PA during pregnancy on the incidence of GDM and reported that, in comparison to the control group, the exercise intervention was shown to significantly reduce the risk of developing GDM the evidence was considered to be high using the Jadad scale [29]. However, it did not affect birth weight, gestational age at birth, the 2 h oral glucose tolerance test (OGTT), or preterm birth [29]. According to another meta-analysis by Aune et al., increased leisure-time PA before, during, and combined (before and during pregnancy) reduced the relative risk of GDM [22]. Higher total PA before pregnancy was associated with a 34% reduction in the relative risk of gestational diabetes, whereas the association for total PA during pregnancy was in the direction of a reduced risk but was not statistically significant, possibly due to the small number of studies [22]. The highest reduction in the relative risk was that of leisure time PA before and also during the pregnancy, which was 59%; the evidence was considered to be moderate to high quality using the Newcastle–Ottawa scale [22].

Physiological IR may mitigate the benefits of exercise during pregnancy, which might explain the lower reduction in relative risk in interventions during that period. In that sense, prenatal exercise and early-in-pregnancy intervention programs are shown to be more preventive. Davenport et al. [24] measured the effect of prenatal exercise as well as the effect of exercise accompanied by dietary interventions on GDM risk. Moderate-quality to high-quality evidence showed that exercise interventions reduced the risk of developing GDM, gestational hypertension (GH), and pulmonary embolisms (PE) by 38%, 39%, and 41%, respectively [24]. The sub-analysis revealed that pregnant women must have at least 600 metabolic equivalents-min/week (MET-min/week) of moderate-intensity exercise to reduce their chances of developing GDM, PE, and GH by 25% (e.g., 140 min of brisk walking, water aerobics, stationary cycling, or resistance training) [24]. Moreover, a review article of 11 randomized controlled trials (RCTs) by Brown et al. has demonstrated that when compared to control (standard treatment), there was a reduction in both fasting blood glucose (FBG) (standardized mean difference (SMD) −0.59, 95% CI −1.07 to −0.11) and in postprandial blood glucose concentration (SMD −0.85, 95% CI −1.15 to −0.55). Furthermore, there was also a trend toward less insulin therapy (OR: 0.59, 95% CI: 0.28–1.22) [35].

Da Silva et al. conducted a meta-analysis of 30 RCTs that measured the effect of participation in leisure-time PA during pregnancy. Results have indicated a reduction in weight gain during pregnancy (−1.11 kg), a lower risk of GDM, and a lower chance of delivering a large for gestational-age newborn, but the evidence was considered to be of low quality using the Jadad Scale to assess outcomes [23]. More precisely, Mijatovic-Vukas et al. [25] associated different types of PA interventions with the risk of GDM and found that GDM risk decreased when engaging in any type of PA early in pregnancy by 21% and when engaging in any type of PA pre-pregnancy by 30% [25]. As well, GDM risk was decreased by 46% when engaging in >90 min/week pre-pregnancy and by 48% when engaging in >15 MET h/week pre-pregnancy [25]. Along with the previous results, a meta-analysis by Russo et al. [28] included 10 RCTs, measuring the effect of the initiation of PA by gestational age (GA). The majority of pregnant women were at <16 weeks of gestation. The authors concluded that exercise intervention during early pregnancy can slightly reduce the risk of GDM by 28% when done through a group exercise plan or an individualized plan [28]. A similar study was conducted by Doi SA et al. [34] in a meta-analysis of 11 RCTs, where it revealed that PA administered in a facility and initiated before the 16–20th week of gestation can prevent the development of GDM in high-risk pregnant women [34].

In another meta-analysis by Ming et al., the studies included were categorized by the different diagnostic criteria of GDM overall, and it was found that PA reduces the risk of GDM by 40–42% and decreases the GWG by 1.61 kg [26]. Furthermore, a meta-analysis conducted by Nasiri-Amiri et al. [27] found that PA interventions conducted during pregnancy were effective in reducing GDM risk by 24%, while engaging in PA 3 times per week was effective in reducing the risk of GDM by 41%; the evidence was considered to be of moderate-to-high quality using the Borge-Scale to assess outcomes [27]. However, no significant impact on the risk of GDM was reported during the first and second trimesters, possibly due to the small number of studies [27]. In the meta-analysis results of Zheng et al. [30], exercise intervention was associated with a significantly reduced incidence of GDM when compared to controls; the evidence was considered high quality evidence using the Jadad Score. However, it did not have any significant impact on preterm birth, gestational age at birth, 2 h OGTT, birth weight, or preeclampsia [30]. On the other hand, a meta-analysis conducted by Madhuvrata P. et al. of 3 RCTS and a total of 183 pregnant women, with a mean age of 30.4 and 30.5 years in the exercise and standard care groups, respectively, and a mean BMI of 34.05 kg/m^2^ in the exercise group vs. 34.5 kg/m^2^ in the standard care group, concluded that in terms of GDM risk, overall, there was no statistical difference between the intervention and control group [31].

Overall, current data shows that PA before and throughout the early stages of pregnancy is crucial for the prevention of GDM [22,25,34,36]. In intervention trials, those who began PA early in pregnancy (<20 weeks of gestation) and those who were organized (often in a group, with an activity leader, e.g., in a health care facility) were the most successful in reducing the risk of developing GDM, particularly when adherence and compliance were high [34]. Few studies have demonstrated a dose-response relationship [25,36], and a few studies have evaluated the effects of different frequencies, intensities, durations, or kinds of exercise on GDM risk [24,36].

## 5. Current Nutrition Therapy for GDM

Nutritional therapy refers to daily meal planning that comprises the provision of energy, macronutrients, and vitamin supplements to guarantee adequate maternal and fetal nutrition while achieving glycemic objectives. A 2017 systematic review from the Cochrane Database including 19 trials, indicated a possible reduction in cesarean section deliveries for women who followed the Dietary Approaches to Stop Hypertension (DASH) versus a control diet at an overall unclear to moderate risk of bias (10 comparisons), but the evidence was considered to be of low quality using GRADE to assess outcomes [37]. Several dietary advice interventions have been studied, such as low-to-moderate glycemic index (GI) diets versus moderate-to-high GI, energy-restricted diets versus non-energy-restricted diets, DASH versus control diets, and low-carbohydrate versus high-carbohydrate diets, rendering challenges in combining the findings to guide clinical practice towards a specific dietary pattern.

The ADA recommends a minimum of 175 g of carbohydrate, 71 g of protein, and 28 g of fiber for all pregnant women [20]. Moreover, it emphasizes the importance of prioritizing monounsaturated and polyunsaturated fats in the diet while avoiding saturated and trans fats. In line with nutrition therapy guidelines for patients with diabetes, both the amount and kind of carbohydrates affect glucose levels [20]. However, liberalizing higher-quality, nutrient-dense carbs was found to result in reduced fasting/postprandial glucose, decreased free fatty acids, improved insulin action, and vascular advantages, as well as the potential reduction of excess infant adiposity [20].

At the present time, there is insufficient data to conclusively support any certain type of dietary regime [37,38,39]. Nonetheless, the Academy of Nutrition and Dietetics evidence-based nutrition practice guideline recommends a customized meal plan that distributes total daily carbohydrate intake over three main meals and two or more snacks, with each meal separated by at least two, and no more than 12 h, between meals for pregnant women with a healthy BMI [40]. Other research advises overweight and obese women to follow a calorie-restricted diet to promote healthy prenatal weight gain while also advising the inclusion of adequate calories and carbohydrates in daily meals to avoid maternal ketosis, which could harm the fetus [41,42].

## 6. Effect of Dietary Interventions on the Prevention of GDM

Both the quantity and the quality of food consumed play a significant role in maintaining glucose homeostasis and improving IR [43]. Even in the absence of excessive caloric consumption, poor diet quality has been demonstrated to have significant long-term effects on beta cell function [44]. Poor diet quality may also result in chronic IR and metabolic dysregulation, with subsequent implications on the likelihood of developing impaired glucose tolerance and T2DM [45]. Using data from the National Health and Nutrition Examination Survey (NHANES), a study conducted in 2015 indicated that a lower diet quality index was more evident in women with a history of GDM compared to their counterparts [46]. Therefore, efforts to prevent GDM should be targeted towards improving diet quality in addition to the usual calorie intake management approach. Similarly, the impact of such strategies was investigated in controlling excessive GWG. A meta-analysis by Rogozińska et al., including 20 RCTs, investigated the effects of several types of interventions, including dietary interventions, combined lifestyle interventions, and supplement use, on the risk of GDM. Six RCTs investigating dietary-based interventions found that a low-GI diet and calorie restriction can reduce the risk of GDM by 33% but do not significantly affect GWG. However, in the same study, a sub-analysis of three studies revealed a statistically significant difference according to BMI for diet-based interventions (*p* = 0.04), as obese and overweight women were significantly less likely to develop GDM (RR 0.40; 95% CI 0.18 to 0.86) [47]. The evidence from this study was considered to be of low quality using GRADE to assess outcomes with a low risk of bias. In another meta-analysis by Bennett et al. [48], which studied the effect of dietary interventions, PA interventions, and combined lifestyle interventions, the results of nine diet-based interventions led to a 44% reduction in the incidence of GDM, indicating that interventions to promote proper GWG can be successful in reducing GDM risks among pregnant women; however, the PA and combined interventions were not significant in reducing the risk of GDM [48]. Rogoziska et al. [47] and Bennett et al. [48] conducted meta-analyses that differed when examining GDM risk stratified by BMI; however, it is important to note that, compared with Rogoziska et al. [47], Bennett et al. included intervention trials that focused on appropriate GWG as the primary outcome along with prevention of GDM. These findings provide an opportunity to evaluate the effectiveness of preconception and/or early pregnancy therapies in women with a high pre-pregnancy BMI.

Several observational studies have revealed that food habits before and throughout pregnancy affect the risk of GDM [49,50,51,52,53]. Furthermore, a study published by the Cochrane Database of Systematic Reviews included 11 trials, with an overall unclear to moderate risk of bias [54], included a review of RCTs and quasi-RCTs that studied the effects of dietary advice versus standard care. Findings indicated a trend toward a decrease in GDM after the dietary intervention when compared to standard care [54]. Overall, among the approaches reviewed, diet-based interventions appear to have a prominent role in the prevention of GDM, but the evidence was considered to be low to very low quality using GRADE to assess outcomes. This might be related to individual dietary and component differences, variations in GDM, and nutritional supplement effects.

## 7. Effect of Combined Interventions (Physical Activity and Dietary Interventions) on the Risk of GDM

A Cochrane Systematic Review by Martis et al. demonstrated that, in addition to glycemic control, self-monitoring of blood sugar levels was linked to decreased GWG during pregnancy and decreased risk of large-for-gestational-age infants (RR: 0.60, 95% CI: 0.50–0.71) [38]. However, the evidence for lifestyle intervention vs. routine care was unclear for long-term outcomes, including hypoglycemia in newborns and childhood obesity [38]. In addition, according to another Cochrane systematic review of 23 RCTs that assessed the effects of combining dietary changes and PA interventions on GDM risk, it showed moderate evidence for a possible reduced risk of GDM with lifestyle interventions when compared with standard care (RR: 0.85, 95%, (CI): 0.71–1.01) [55]. The RCTs included in this review were largely focused on preventing excess GWG, although they differed substantially in design, intervention type, intensity, mode of delivery, and continuity of care [55].

Several meta-analyses and multicenter RCTs have proven the effectiveness of lifestyle counseling for women at high risk of GDM [36,55,56,57,58,59,60,61]. In the Finnish Gestational Diabetes Prevention Study (RADIEL) RCT, prevention of GDM was assessed in pregnant women with a high risk for GDM [57]. Upon enrollment at 20 weeks of pregnancy, 293 women with a history of GDM and/or a pre-pregnancy BMI of 30 kg/m^2^ were randomly assigned to the intervention group (*n* = 155) or the control group (*n* = 138). Each participant in the intervention group got customized counseling on dietary modifications, PA, and weight management from certified nurses and participated in one group session with a dietitian. On the other hand, the control group was provided with standard prenatal care [57]. The results have shown that the intervention group had a 13.9% incidence of GDM compared to 21.6% in the control group (95% CI 0.40–0.98%; *p* = 0.044) [57]. Moreover, the GWG has been observed to be lower in the intervention group compared to the control group by 0.58 kg (95% CI 21.12 to 20.04 kg) [57]. However, in contrast to the RADIEL study, the incidence of GDM did not differ between the similar intervention and control groups in the LIMIT study that was conducted in Australia [59].

Similarly, the Australian HeLPher study examined the impact of lifestyle interventions on a sample of overweight and obese pregnant women. Although the intervention did not have any effect on the prevalence of GDM, a significant difference between the control and intervention groups was observed concerning GWG, where those in the control group gained more weight compared to the intervention group [6.9 (3.3) vs. 6.0 (2.8) kg, *p* < 0.05] [56]. Furthermore, the DALI lifestyle study, which was conducted in nine different European countries [58], measured the effect of vitamin D supplementation as a prevention tool in addition to lifestyle interventions. The results indicated that by 35 to 37 weeks, pregnant women who received the intervention had significantly lower GWG than controls (22.02; 95% CI: 23.58 to 20.08 kg); however, it was concluded that it did not affect glycemic control [58].

Four meta-analyses reviewed the effect of combined lifestyle interventions on the risk of GDM [36,55,60,61]. Bain E et al. [60] concluded that a combination of lifestyle interventions versus standard management had less impact on the risk of developing GDM; however, women who received the dietary and PA advice had lower GWG during pregnancy [60]. Similarly, Song et al. [61] concluded that the GDM risk was 18% lower in pregnant women who received the intervention. Interestingly, subgroup analysis revealed that such an intervention was successful among women who received it before the 15th gestational week [61]. Moreover, Shepherd et al. [55] and Guo et al. [36] had similar results, as they found that GDM was slightly prevented by lifestyle interventions by 15% and 23%, respectively, but the difference was not significant, and the evidence was considered to be low to moderate using GRADE to assess outcomes with an overall unclear to moderate risk of bias [36,55]. Table 2 summarizes the outcomes of these meta-analyses.

## 8. The Effect of Dietary Supplements on the Risk of GDM

### 8.1. Vitamin D

Deficiency of vitamin D is widespread in pregnant women and has been associated with pregnancy complications such as GDM, hypertension, premature birth, and a small gestational age [62]. In women with GDM, administration of vitamin D (1000–4762 IU/day) enhanced glucose metabolism by reducing FBG, HbA1c, and serum insulin concentrations [63]. In the SCOPE study conducted in Australia and New Zealand, where they measured the correlation between levels of vitamin D in pregnant women at 15 ± 1 weeks’ of gestation and the risk of GDM, it was found that pregnant women who had high levels of vitamin D (>81 nmol/L) were protected against GDM [64].

Available evidence suggests that, in pregnant women, vitamin D supplementation significantly reduced FBG (MD = −1.87, 95% CI-3.39–0.35) and the incidence of GDM (OR = 0.42, 95% CI 0.30–0.60) [65]. A meta-analysis published in 2020, found that vitamin D supplementation substantially lowered FBG (−12.13 mg/dL, 95% CI: −14.55 to −9.70) and controlled HOMA-IR, and that this was superior than omega-3 (−3.64 mg/dL, 95% CI: −5.77 to −1.51), zinc (−5.71 mg/dL, 95% CI: −10.19 to −1.23), and probiotics (−6.76 mg/dL, 95% CI: −10.02 to −3.50) for improving FBG [66]. The impact of vitamin D supplementation in combination with other supplements has also been investigated in patients with GDM. Apart from enhancing IR indices, a dose of 1000 IU/day of vitamin D3 lowered serum triglycerides, VLDL, total cholesterol, and LDL concentrations considerably [67]. More recently, 200 IU of vitamin D, 100 mg of magnesium, 4 mg of zinc, and 400 mg of calcium co-supplementation for six weeks in women with GDM reduced inflammation and oxidative stress biomarkers (CRP and plasma malondialdehyde concentrations) and increased total antioxidant capacity levels [68]. Furthermore, the supplementation showed a declining trend in birth weight and macrosomia rate (3.3 vs. 16.7%, *p* = 0.08) [68].

### 8.2. Myo-Inositol

Inositol, a naturally occurring polyol sugar, is found in grains, beans, nuts, meat, legumes, and fresh citrus fruits. It is an insulin-sensitizing mediator that has been shown to increase insulin sensitivity and ovulatory function in young women with polycystic ovarian syndrome (PCOS).

Pregnant women with a family history of T2DM were included in a prospective RCT, excluding patients who were obese or had a history of PCOS, GDM, or pre-gestational diabetes. Participants were randomized to either (2.0 g) myo-inositol plus 200 mcg folic acid twice a day or 200 mcg folic acid twice a day. By the end of the first trimester, myo-inositol supplementation decreased the incidence of GDM (6.0 vs. 15.3%, *p* = 0.04) and macrosomia [69]. Moreover, in another RCT among obese pregnant women, when taken at the end of the first trimester, myo-inositol (2.0 g) also decreased the incidence of GDM in the intervention group compared to the placebo group (33.6% vs. 14.0%) [70]. Furthermore, a more recent systematic review and meta-analysis of 5 RCTs found that myo-inositol (2.0 g) was associated with a substantial decrease in the incidence of GDM and preterm birth [71]. Therefore, the possible advantage of myo-inositol in enhancing insulin sensitivity implies that it could help prevent GDM in obese or overweight women, women with PCOS, or women with a family history of T2DM. However, extensive RCTs in multiple ethnic groups and more detailed risk factor profiles are needed before myo-inositol may be suggested to prevent GDM.

## 9. Effect of Combined Interventions (Physical Activity and Dietary Interventions) on Outcomes in Patients with GDM

### 9.1. Glycemic Control

A meta-analysis of four trials showed that lifestyle interventions were associated with a decrease in postprandial blood glucose (MD: −27.11 mg/dL; 95% CI −44.62–9.61) [72]. When measuring the effect of lifestyle interventions on HbA1c, there was a reduction in HbA1c MD −0.33 mmol/mol; 95% CI −0.47 to −0.19) in pregnant women with GDM at the end of the lifestyle intervention treatment; however, no effect on FBG was observed following the treatment [72].

### 9.2. Weight Gain during Pregnancy

A meta-analysis of 4 RCTs found that GWG was reduced by lifestyle interventions (MD −1.30 kg, 95% CI −2.26 −0.35) [72]. Moreover, in a RCT by Landon et al. [73], the greatest difference between the intervention and control groups was recorded (2 kg in the lifestyle intervention group versus 5 kg in the control group).

### 9.3. Postnatal Depression

Crowther et al. [74] recorded the effect of lifestyle interventions on postnatal depression, which was characterized as an Edinburgh Postnatal Depression Score greater than 12. When compared to the control group, lifestyle interventions were associated with a lower incidence of postnatal depression (RR 0.49, 95% CI 0.31–0.78) [74].

## 10. Future Directions

Individual RCTs and meta-analyses demonstrate the urgent need to transition from the “one-size-fits-all” approach of dietary and PA recommendations during pregnancy to a personalized approach that considers individual-level risk factors, behaviors, and socioeconomic status. Although the RCTs included in the previously disclosed meta-analyses had varying approaches, the majority were based on pre-defined dietary and PA guidelines. Dietary [40] and GWG recommendations [75] have been altered to provide more specific advice on energy requirements and weight gain by considering both BMI classification and trimester since some individuals may require fewer lifestyle alterations to attain excellent pregnancy results. Considering the uniqueness of each pregnancy and the outcome of recent RCTs, it is evident that there is uncertainty regarding the most beneficial dietary and PA habits. Effective treatments require an understanding of the risk at the individual level; therefore, early prenatal screening is required to determine if a woman has a low, moderate, or high risk of having GDM. Risk prediction systems should consider a broad spectrum of clinical features and physiological measurements, such as glucose metabolism and insulin sensitivity, and integrate new biomarkers from metabolomic, genetic, and microbiome investigations. Although there is insufficient evidence to support screening and treatment for GDM during the early stages of pregnancy, effective algorithms that use early pregnancy risk factors to predict who may develop GDM, will help target approaches to reduce the risk of GDM development and postpartum associated complications. These risk prediction algorithms can be used with RCTs of lifestyle interventions. Together, physicians may use these findings to assist patients based on their risk profiles, by using various intervention strategies.

While there is some evidence that lifestyle interventions are beneficial, there are still significant knowledge gaps that make it difficult to guide clinical practice. This is especially true when it comes to identifying the specific aspects of lifestyle interventions, such as dietary changes and exercise, that reduce the risk of or improve GDM in pregnant women. Evidence from the quality of the meta-analyses, which shows a low to moderate quality and a moderate to high risk of bias, emphasizes the importance of conducting extensive, high quality, large-scale studies on the causes and remedies for GDM and harmonizing clinical practice.

## 11. Conclusions

This evidence-based review of the current literature on the effectiveness of lifestyle interventions for preventing and treating GDM highlighted several important research gaps which need to be further ascertained. Various lifestyle interventions have a positive effect on GDM prevention. The most effective PA intervention for lowering the risk of developing GDM are those delivered early in pregnancy (>20 weeks of gestation) and those who were organized in a healthcare institution (typically in a group). Moreover, dietary interventions that targeted diet quality were effective in the prevention of GDM early in pregnancy and in controlling GWG throughout the pregnancy. Additionally, supplementing with vitamin D can improve FBG and HOMA-IR; however, no optimal dose can be identified based on the wide range used in the above-mentioned studies (200-4762 IU). Myo-inositol supplementation improves insulin sensitivity in obese or overweight women, as well as in women with PCOS or a family history of T2DM, and helps prevent GDM in doses of 2.0 g. A brief overview of the literature on the effect of combined interventions on GDM outcome indicates a positive effect on glycemic control in pregnancy, reduction of GWG, and the incidence of postnatal depression.

## Figures and Tables

**Figure 1 medicina-59-00287-f001:**
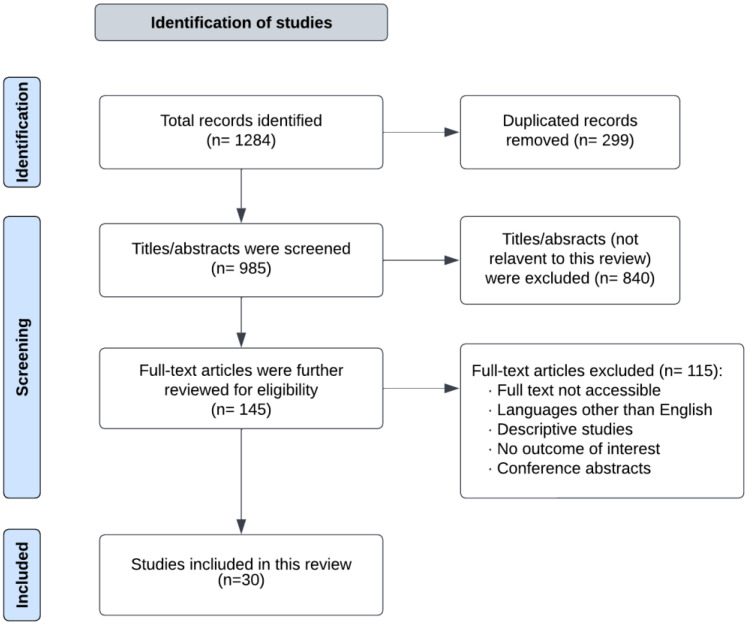
Flow chart of included and excluded studies.

**Table 1 medicina-59-00287-t001:** Findings of meta-analyses measuring the effect of physical activity on the risk of GDM.

Author, Year	Studies Included	Number of Participants	Intervention	Outcome	Effect	95% CI	Risk Reduction	Quality of the Evidence (GRADE)
Ying Yu et al., 2018 [29]	5 RCTs	1370	Exercise during pregnancy	A physical activity intervention might substantially reduce the incidence of gestational diabetes mellitus.	OR: 0.59	CI = 0.39 to 0.88	41%	NA
Aune D et al., 2016 [22]		4607	Low vs. high-intensity physical activity before pregnancy	High-intensity training before pregnancy was more effective in reducing GDM risk	RR: 0.62	CI = 0.41 to 0.94	38%	NA
	3 cohorts	3996	Low vs. high-intensity physical activity during pregnancy	High-intensity training during pregnancy was more effective in reducing GDM risk	RR: 0.66	CI = 0.36 to 1.21	34%	NA
	2 cohorts	2714	Physical exercise in leisure time before and during pregnancy	Compared to women who were sedentary before and throughout pregnancy, women who were physically active before and during pregnancy had a 59% lower relative risk.	RR: 0.41	CI = 0.23 to 0.73	59%	NA
Davenport MH et al., 2018 [24]	26 RCTs	6934	Prenatal exercise compared to no exercise	Exercise-only therapies decreased the likelihood of developing GDM by 38% compared to those that did not involve exercise.	OR: 0.62	CI = 0.52 to 0.75	38%	Moderate
Da Silva SG. et al., 2017 [23]	10 RCTs	3790	Leisure-Time Physical Activity	Preventive effect of leisure-time physical activity interventions on the onset of GDM	RR: 0.67	CI = 0.49–0.92)	33%	NA
	6 Cohorts	6754			OR: 0.75	CI = 0.55 to 1.01	25%	NA
Mijatovic-Vukas. et al., 2018 [25]	11 RCTs	57,722	Any type of physical activity pre-pregnancy	Pre-pregnancy physical activity reduced the risk of GDM by 30%	OR: 0.70	CI = 0.57 to 0.85	30%	NA
	8 RCTs	23,717	Any type of physical activity early in pregnancy	Early in pregnancy physical activity reduced the risk of GDM by 21%	OR: 0.79	CI = 0.64 to 0.97	21%	NA
	6 RCTs	45,162	>15 MET hr./week pre-pregnancy	Reduced the risk of GDM by 48%	OR: 0.52	CI = 0.27 to 1.00	48%	NA
	4 RCTs	19,730	>90 min/week pre-pregnancy	Reduced risk of GDM by 46%	OR: 0.54	CI = 0.34 to 0.87	46%	NA
Russo LM. et al., 2015 [28]	10 RCTs	3401	Group exercise intervention or individualized plan until GDM screening	Physical exercise during pregnancy has a minor protective impact against gestational diabetes.	RR: 0.72	CI = 0.52–0.97	28%	Low
Ming W-K et al., 2018 [26]	8 RCTs	2981	Physical activity during pregnancy	Positive effect on decreasing the incidence of GDM	Depending on different diagnostic criteria			
					RR: 0.58	CI = 0.37 to 0.90	42%	NA
					RR: 0.60	CI = 0.36 to 0.98	40%	NA
Nasiri-Amiri F et al., 2019 [27]	8 RCTs	1441	Physical activity during pregnancy	Physical activity during pregnancy, in the second trimester and 3 times/week were effective in reducing the risk of GDM by 24%, 36%, and 41% respectively.	RR: 0.76	CI = 0.56 to 1.03	24%	NA
	4 RCTs	1022	Physical activity 3 times/week or less		RR: 0.59	CI = 0.46 to 0.76	41%	NA
	4 RCTs	797	Physical activity in the second trimester		RR: 0.64	CI = 0.40 to 1.04	36%	NA
Zheng J. et al., 2017 [30]	5 RCTs	1872	Exercise at 10–22 weeks of gestation	An exercise intervention has been reported to significantly lower the risk of GDM when compared to a control intervention.	OR: 0.62	CI = 0.43 to 0.89	38%	NA
Madhuvrata P. et al., 2015 [31]	3 RCTs	183	Exercise during pregnancy	In terms of GDM, there was no statistical difference between the two groups.	OR: 0.77	CI = 0.33 to 1.79	23%	NA
Doi SA et al., 2020 [34]	11 RCTs	1467	Physical activity was delivered in healthcare facilities to pregnant women in high-risk pregnancies before the 20th week of gestation.	GDM risk was found to be significantly lower in pregnant women in the intervention group	RR: 0.53	CI = 0.38 to 0.74	49%	NA

Abbreviations: OR: Odds Ratio; RR: Relative Risk; CI: Confidence Interval; RCTs: Randomized Controlled Trials; GDM: Gestational Diabetes Mellitus; MET: metabolic equivalents; NA: Not Applicable.

**Table 2 medicina-59-00287-t002:** Summary of meta-analysis on the effect of lifestyle interventions (combination of dietary changes and physical activity).

Author, Year	Studies Included	Number of Participants	Groups	Outcome	Relative Risk	Quality of the Evidence (GRADE)
Bain E et al., 2015 [60]	13 RCTs	4983	Dietary and physical activity combination lifestyle interventions vs. standard management	There was no discernible difference in the risk of developing GDM Women who received the combined diet and activity intervention had lower GWG during pregnancy (mean difference (MD) 0.76 kg, (CI 1.55 to 0.03)	RR: 0.92, 95% (CI 0.68 to 1.23)	Moderate
Song et al., 2016 [61]	29 RCTs	11,487	Dietary, physical activity, or combination lifestyle interventions vs. standard management	GDM risk was reduced by 18% (95 percent CI 5–30%) (*p* = 0.0091). Subgroup analysis revealed that such an intervention was successful among women who received it before the 15th gestational week	RR: 0.80, 95% CI 0.66–0.97).	Low to moderate
Shepherd et al., 2017 [55]	23 RCTs	8918	Combination of dietary and physical activity interventions vs. standard management	The diet and exercise intervention group had a probable lower risk of GDM compared to the standard care group	RR: 0.85, 95% (CI 0.71 to 1.01)	Moderate
Guo et al., 2019 [36]	47 RCTs	15,745	Dietary, physical activity, or combination lifestyle interventions vs. standard management	GDM was prevented by diet and physical activity during pregnancy	RR: 0.77, 95% (CI 0.69–0.87)	Low

Abbreviations: RR: Relative Risk; CI: Confidence Interval; RCTs: Randomized Controlled Trials; GDM: Gestational Diabetes Mellitus.

## Data Availability

Not applicable.

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
