# Peer review of "The Role of Lifestyle Interventions in the Prevention and Treatment of Gestational Diabetes Mellitus"

_medicina, 2023, doi:10.3390/medicina59020287_

Round 1

Reviewer 1 Report

See attached

Author Response

Response to Reviewer 1 Comments

Point 1: This review paper gives a comprehensive summary of the effect lifestyle interventions (physical activity and dietary interventions) have in the prevention and treatment of gestational diabetes mellitus (GDM). It reviews the current physical activity and nutrition recommendations and captures what their effect is on prevention and outcomes of GDM based on evidence reported in systematic reviews and meta-analysis (SR-MA). Overall, the content is clear and easy to follow. The manuscript is well written.

Response 1: Thank you so much.

 General comments:

Point 2: Regarding the body of the manuscript, the methodology is absent. Given that the aim of the review is to summarize and assess current evidence on the role of lifestyle interventions in the prevention and treatment of GDM the manuscript would greatly benefit from reporting: what type of journals were searched? What years? What key term were used? It is clear that this is not a systematic review but providing this information would set the bases of the scope of the manuscript.

Response 2: Thank you. The methodology section has been added as requested. As well as a flowchart of the methodology process.

Point 3: Furthermore, how was the evidence/quality from the included SR-MAs assessed? The aim of the study was to “assess current evidence” but it is not clear how this was done. This needs to be specified. It would be of interest to address both, the quality (Risk of bias) as well as the certainty of the evidence (GRADE) of the SR-MAs included in the review. As authors stated, even though there is some evidence of the benefits that lifestyle interventions have on GDM there is a need for future studies. Addressing how confident the overall estimate of the effect of the SR-MA present would provide a stronger basis for their conclusion. So, for example if most SR-MA reported a low to moderate GRADE then this would greatly support their conclusion, however if they are reporting a high GRADE then future research is very unlikely to change the confidence in the estimate of effect and the conclusion is not supported. Reporting on the “assessment” part of the SR-MA is of relevance and interest to the scientific and general community.

Response 3: Thank you for your great suggestion. The risk of bias and the quality of the evidence (GRADE) have been added to SR-MA were applicable.

Specific comments:

Point 4: Line 27: Abstract: The term “Precision prevention in mind” is used but in the body of the manuscript this term in not mentioned. It would be recommended either to remove it or talk about it within the discussion using the same terms.

Response 4: Thank you. Noted and updated.

Point 5: Line 34-65: Introduction: The introduction would benefit from including background on the emerging field of the developmental origins of health and disease (DOHaD) which puts forward the idea that intrauterine environments have a permanent conditioning and/or programming on metabolism and health later in life. Furthermore, review more in detail the metabolic benefits of physical activity and nutrition interventions.

These are nice manuscripts that will strengthen the introduction

  • Fu J, Retnakaran R. The life course perspective of gestational diabetes: An opportunity for the prevention of diabetes and heart disease in women. EClinicalMedicine. 2022 Feb 12;45:101294.
  • McIntyre, H.D., Catalano, P., Zhang, C. et al. Gestational diabetes mellitus. Nat Rev Dis Primers 5, 47 (2019). https://doi.org/10.1038/s41572-019-0098-8

Response 5: Thank you for your comment and suggestion. The introduction has been updated. 

Point 6: Table 1: Ying Yu et al, 95% CI has a “vealed” that needs to be removed.

Response 6: Noted and removed.

Point 7: Line 95: There is Ref missing. Line 152: Double comma, remove.

Response 7: Noted and removed.

Point 8: Line 205-206: Not very clear, please rephrase sentence. Line 234: Add “by” before Martis et al.,

Response 8: Thank you. Noted and updated.

Point 9: Table 2: In the outcomes of Bain E. et al., there is a “0.76 kg,%” correct.

Response 9: Thank you. Noted and corrected.

Point 10: Line 290: FBG abbreviation meaning missing. The full name is used later in line 296 but the abbreaviation is not use. Please correct

Response 10: Thank you. Noted and corrected.

Point 11: Line 285 – 309: Vit D: Please include the dosses used. This is important information to report to know at what dosses these effects where seen.

Response 11: Thank you. Noted and updated.

Point 12: Line 381 – 390: Conclusion: It would benefit from including a detailed summary of the main finding of each area. It seems like specifically for physical activity interventions, there is a time frame (before 16-20 wks) where it is more effective and that having structured supervised also provides the greatest benefits however the type of physical activity does not appear to be critical. For nutrition intervention focusing more on the quality seems to plays a role. So, providing a summary of all this finding would make the conclusion stronger and more informative.

Response 12: Thank you. The conclusion has been updated and comments were addressed.

Point 13: Line 53, 72, 117, 142, 144, 166, 176, 240, etc…) throughout the manuscript there are various double spaces. Please address them.

Response 13: Thank you for noting that. Addressed.

Reviewer 2 Report

It is a narrative review study. Broadly speaking, it is well-written and easy to read. It has a few editing flaws, which are presented below.

Table 1.

a)     The title of the column “meta-analysis” is wrong. The information corresponds to the source.

b)     What does “mean difference” mean? Is it an odds ratio or relative risk? Is the 95%CI also averaged?

c)      What does RR mean? What is the difference between the column “mean difference” and the column “RR”?

d)     Row 1, column 95%CI, there is a text that does not correspond “vealed”

Line 95. The text [Ref] appears without the corresponding reference number

Table 2.

a)     The title of the column “meta-analysis” is wrong. The information corresponds to the source.

b)     Does “RR” mean “risk reduction” or “relative risk”? (They are not synonymous, although it is understood that a relative risk of less than 1 reduces the risk, this is an interpretation of the result)

Author Response

Response to Reviewer 2 Comments

Point 1: It is a narrative review study. Broadly speaking, it is well-written and easy to read. It has a few editing flaws, which are presented below.

Response 1: Thank you so much.

Point 2: Table 1.

a)The title of the column “meta-analysis” is wrong. The information corresponds to the source.

Response 2: Thank you. Noted and corrected.

Point 3: b)What does “mean difference” mean? Is it an odds ratio or relative risk? Is the 95%CI also averaged?

Response 3: Thank you. The mean difference in this table means either Odds Ratio or Relative Risk, depending on the type of studies used in the meta-analysis. This has been made clear in the table and defined under the column “effect” of each meta-analysis whether it is OR or RR.

Point 4: c)What does RR mean? What is the difference between the column “mean difference” and the column “RR”?

Response 4: Thank you. RR in this table means the Risk Reduction (RR). RR was calculated from the Odds ratio or the Relative Risk. This has been clarified in the footnote of the table.

Point 5: d)Row 1, column 95% CI, there is a text that does not correspond “vealed”

Response 5: Thank you. Noted and corrected.

Point 6: Line 95. The text [Ref] appears without the corresponding reference number

Response 6: Thank you. Noted and corrected.

Point 7: Table 2.

a)The title of the column “meta-analysis” is wrong. The information corresponds to the source.

Response 7: Thank you. Noted and corrected.

Point 8: b) Does “RR” mean “risk reduction” or “relative risk”? (They are not synonymous, although it is understood that a relative risk of less than 1 reduces the risk, this is an interpretation of the result)

Response 8: Thank you. In table 2 RR indicates Relative Risk. Noted and added as an abbreviation under the table.

Round 2

Reviewer 1 Report

- Please add the aim of the review paper to the end of the introduction. 

- Review figure 1 as the numbers don't add up. If 299 duplicates were removed that would leave 985 titles to screen not 994, please correct this.

- The authors' conclusion is too long. Authors could use the information provided as part of the discussion section and work on a concise conclusion (one paragraph). 

Line 460 - Is sample size really a limitation factor? Was there no sample size calculation provided in these studies? If there was then increasing the sample size would not have an effect and is not a limiting factor. 

Line 434 - an "are" is missing (developing GDM are interventions...)

Line 440 - remove "-" in im-prove

Author Response

Response to Reviewer 1 Comments- Round 2

Point 1: Please add the aim of the review paper to the end of the introduction.

Response 1: Thank you. Noted and added.

Point 2: Review figure 1 as the numbers don't add up. If 299 duplicates were removed that would leave 985 titles to screen not 994, please correct this.

Response 2: Thank you. Noted and corrected.

Point 3: The authors' conclusion is too long. Authors could use the information provided as part of the discussion section and work on a concise conclusion (one paragraph).

Response 3: Thank you. Noted and addressed.

Point 4: Line 460 - Is sample size really a limitation factor? Was there no sample size calculation provided in these studies? If there was then increasing the sample size would not have an effect and is not a limiting factor.

Response 4: Thank you for your comment. This has been checked and edited from the text as required.

Point 5: Line 434 - an "are" is missing (developing GDM are interventions...)

Response 5: Thank you. Corrected.

Point 6: Line 440 - remove "-" in im-prove

Response 6: Thank you. Corrected.
